# SPACA6 ectodomain structure reveals a conserved superfamily of gamete fusion-associated proteins

Tyler D. R. Vance [1], Patrick Yip[1], Elisabet Jiménez [2], Sheng Li [3], Diana Gawol[1], James Byrnes[4], Isabel Usón[2,5], Ahmed Ziyyat [6,7] & Jeffrey E. Lee [1✉]

SPACA6 is a sperm-expressed surface protein that is critical for gamete fusion during mammalian sexual reproduction. Despite this fundamental role, little is known about how SPACA6 specifically functions. We elucidated the crystal structure of the SPACA6 ectodomain at 2.2-Å resolution, revealing a two-domain protein containing a four-helix bundle and Ig-like β-sandwich connected via a quasi-flexible linker. This structure is reminiscent of IZUMO1, another gamete fusion-associated protein, making SPACA6 and IZUMO1 founding members of a superfamily of fertilization-associated proteins, herein dubbed the IST superfamily. The IST superfamily is defined structurally by its distorted four-helix bundle and a pair of disulfide-bonded CXXC motifs. A structure-based search of the AlphaFold human proteome identified more protein members to this superfamily; remarkably, many of these proteins are linked to gamete fusion. The SPACA6 structure and its connection to other IST-superfamily members provide a missing link in our knowledge of mammalian gamete fusion.

[1] Department of Laboratory Medicine and Pathobiology, Temerty Faculty of Medicine, University of Toronto, Toronto, ON, Canada. [2] Institute of Molecular Biology of Barcelona (IBMB-CSIC), 08028 Barcelona, Spain. [3] Department of Medicine, University of California San Diego, La Jolla, CA, USA. [4] National Synchrotron Light Source II, Brookhaven National Laboratory, Upton, NY, USA. [5] ICREA, Pg. Lluís Companys 23, 08010 Barcelona, Spain. [6] Université Paris Cité, CNRS, INSERM, Institut Cochin, F-75014 Paris, France. [7] Service d'Histologie, d'Embryologie, Biologie de la Reproduction, AP-HP, Hôpital Cochin, F-75014 Paris, France. ✉email: jeff.lee@utoronto.ca

Every human life begins with two separate haploid gametes: a sperm from the father and an oocyte from the mother. This sperm was the winner of an intensive selection process in which millions of sperm traversed the female reproductive tract, passed through various barriers[1], and underwent capacitation processes that augmented their motility and surface composition[2–4]. Even once the sperm and oocyte have found each other, the process is not over. Oocytes are surrounded by a layer of cumulus cells as well as a glycoprotein barrier called the zona pellucida, both of which the sperm must pass through to gain access to the oocyte. Sperm use a combination of surface adhesion molecules and membrane-associated and secreted enzymes to break through these final barriers[5]. These molecules and enzymes are predominantly stored within the inner membrane and acrosome matrix and are revealed through the dissolution of the outer membrane of the sperm during the acrosome reaction[6]. The final step of this intensive journey is the sperm-egg fusion event, where the two cells merge their cell membranes and become a single diploid organism[7]. Despite this process being a seminal one in human reproduction, little is known about the molecular interactions required.

Outside of gamete fertilization, the chemical process of fusing two lipid bilayers has been studied extensively. In general, membrane fusion is an energetically unfavorable process, requiring protein catalysts that undergo changes in structural conformation to draw two membranes close together, disrupt their continuity, and induce fusion[8,9]. Dubbed fusogens, these protein catalysts have been found in a myriad of fusion systems. They are necessary for viral entry into host cells (e.g., gp160 in HIV-1, spike in coronaviruses, hemagglutinin in influenza viruses)[10–12], the formation of the placenta (syncytins)[13–15], and in gamete fusion of lower eukaryotes (HAP2/GCS1 in plants, protists, and arthropods)[16–19]. The fusogen for human gametes has yet to be discovered, although several proteins have been shown to be vital for gamete attachment and fusion[20]. The first discovered was the oocyte-expressed CD9, a transmembrane protein necessary for gamete fusion in both mice and humans[21–23]. Though its exact function remains unclear, roles in adhesion, structuring of adhesive focal points on egg microvilli, and/or proper localization of oocyte-surface proteins appear likely[24–26]. The two best characterized proteins critical for gamete fusion are the sperm protein IZUMO1[27] and the oocyte protein JUNO[28], which bind each other as an essential step in gamete recognition and pre-fusion adhesion. *Izumo1* knockout male and *Juno* knockout female mice are completely infertile; in these models, sperms penetrate the perivitelline space, but gametes are unable to fuse[27,28]. Likewise, in human in vitro fertilization experiments, there is a reduction in fusion when gametes are treated with antibodies against IZUMO1 or JUNO[27,29].

Recently, a collection of newly discovered, sperm-expressed proteins with similar phenotypes to IZUMO1 and JUNO have been discovered[20,30–35]. Sperm Acrosome Membrane-Associated protein 6 (SPACA6) was identified as essential in fertilization during a large-scale mutagenesis study in mice. Transgene insertion into the *Spaca6* gene produced sperm unable to fuse, although these sperm penetrated into the perivitelline space[36]. Subsequent knockout studies in mice confirmed that *Spaca6* is essential for gamete fusion[30,32]. SPACA6 is expressed almost exclusively in the testis and has a localization pattern similar to that of IZUMO1, i.e. within the inner membrane of sperm prior to the acrosome reaction, followed by relocation to the equatorial region post-acrosome reaction[30,32]. Homologs to *Spaca6* are found in a variety of mammals and other eukaryotes[30], and its importance for human gamete fusion has been established through anti-SPACA6 inhibition of human in vitro fertilization[30]. Unlike IZUMO1 and JUNO, the specifics regarding the structure, interactions, and function of SPACA6 remain unclear.

In the interest of better understanding the fundamental processes behind human sperm-egg fusion – thereby informing future advances in both family planning and infertility treatment – we undertook structural and biochemical studies of SPACA6. The crystal structure of the SPACA6 ectodomain revealed a four-helix bundle (4HB) and immunoglobulin-like (Ig-like) domain that are connected by a quasi-flexible region. As predicted in previous studies[7,32,37], the domain architecture of SPACA6 is similar to that of human IZUMO1, with both proteins sharing an uncommon motif: a 4HB with a triangular face of helices and pair of disulfide-bonded CXXC motifs. We propose that IZUMO1 and SPACA6 now define a larger, structurally related superfamily of gamete fusion-associated proteins. Using the hallmark features specific to the superfamily, we carried out an exhaustive search of the AlphaFold structural human proteome, revealing additional members of this superfamily, including several linked to gamete fusion and/or fertilization. It now appears that there is a common structural fold and superfamily of proteins that are associated with gamete fusion, with our structure providing a molecular picture of this important aspect of the human gamete-fusion machinery.

## Results

**A soluble monomeric SPACA6 ectodomain.** SPACA6 is a single-pass transmembrane protein with one N-linked glycan and six predicted disulfide linkages (Figs. S1a and S2). We expressed the extracellular domain of human SPACA6 (residues 27–246) in *Drosophila* S2 cells and purified the protein using nickel-affinity, cation-exchange, and size-exclusion chromatographies (Fig. S1b). The purified SPACA6 ectodomain was highly stable and homogeneous. Analysis with size-exclusion chromatography coupled multi-angle light scattering (SEC-MALS) revealed a single peak with a calculated molecular weight of 26.2 ± 0.5 kDa (Fig. S1c). This is consistent with the size of a monomeric SPACA6 ectodomain, indicating that no oligomerization occurred during purification. Furthermore, circular dichroism (CD) spectroscopy revealed mixed α/β structure with a melting temperature 51.3 °C (Fig. S1d, e). Deconvolution of the CD spectra showed 38.6% α-helix and 15.8% β-strand elements (Fig. S1d).

**SPACA6 has two domains: a four-helix bundle and an Ig-like β-sandwich.** The SPACA6 ectodomain was crystallized using a random matrix microseeding approach[38], yielding a 2.2-Å resolution dataset (Table 1 and Fig. S3). The structure was determined using a combination of fragment-based molecular replacement and SAD phasing data from bromide soaks (Table 1 and Fig. S4), with the final refined model consisting of residues 27–246. At the time of structure determination, there were no experimental or AlphaFold structures available. The SPACA6 ectodomain, with dimensions of 20 Å × 20 Å × 85 Å, is made up of seven helices and nine β-strands and adopts an elongated tertiary fold stabilized by six disulfide bonds (Fig. 1a, b). Weak electron density at the end of the Asn243 sidechain suggests this residue is N-linked glycosylated. The structure consists of two domains: an N-terminal four-helix bundle (4HB) and a C-terminal Ig-like domain, with an intermediary hinge region between the two (Fig. 1c).

The 4HB domain of SPACA6 includes four main helices (Helices 1–4) arranged in a coiled-coil fashion (Fig. 2a) that alternate between antiparallel and parallel interactions (Fig. 2b). A small additional single-turn helix (Helix 1′) packs perpendicularly with the bundle, forming a triangular shape with Helices 1 and 2. This triangle produces a slight distortion in the coiled-coil packing relative to the tight packing of Helices 3 and 4 (Fig. 2a).

**Table 1 Data collection and refinement statistics.**

| | ARCIMBOLDO SPACA6 | SPACA6 Br-SAD |
|---|---|---|
| Synchrotron | APS 24ID-C (NE-CAT) | APS 24ID-C (NE-CAT) |
| Wavelength (Å) | 1.6314 | 0.91165 |
| Resolution range (Å)[a] | 43.44–2.00 (2.072–2.000) | 44.83–2.25 (2.331–2.25) |
| Space group | $P2_12_12_1$ | $P2_12_12_1$ |
| Unit cell | | |
| a, b, c (Å) | 27.6 44.5 195.9 | 29.0 46.5 167.7 |
| α, β, γ (°) | 90 90 90 | 90 90 90 |
| Total reflections[a] | 57321 (3492) | 82666 (6844) |
| Unique reflections[a] | 16683 (1503) | 11,435 (1107) |
| Multiplicity[a] | 3.4 (3.2) | 7.2 (6.7) |
| Completeness (%)[a] | 96.7 (90.1) | 99.8 (99.9) |
| Mean I/sigma(I)[a] | 8.6 (1.2) | 11.7 (2.8) |
| Wilson B-factor | 37.82 | 34.32 |
| $R_{merge}$ (%)[a,b] | 6.6 (65.4) | 10.8 (71.6) |
| $R_{meas}$ (%)[a,c] | 8.8 (91.8) | 11.6 (81.5) |
| $R_{pim}$ (%)[a,d] | 4.6 (48.5) | 4.1 (30.3) |
| CC1/2[a] | 0.997 (0.675) | 0.998 (0.871) |
| $R_{work}$ (%)[a,e] | 27.6 (29.3) | 20.1 (22.0) |
| $R_{free}$ (%)[a] | 29.8 (30.5)[f] | 25.5 (30.1)[g] |
| Number of non-hydrogen atoms | 1592 | 1811 |
| macromolecules | 1592 | 1736 |
| bromide | N/A | 10 |
| solvent | 0 | 65 |
| RMSD (bonds; Å) | 0.010 | 0.005 |
| RMSD (angles; °) | 1.2 | 1.0 |
| Ramachandran favored (%) | 90.3 | 95.4 |
| Ramachandran allowed (%) | 6.6 | 4.6 |
| Ramachandran outliers (%) | 3.1 | 0.0 |
| Clashscore | 9.6 | 1.8 |
| Average B-factor (Å²) | 44.5 | 44.0 |
| macromolecules | 44.5 | 44.3 |
| bromide | N/A | 50.9 |
| solvent | N/A | 34.4 |
| Number of TLS groups | N/A | 7 |
| PDB accession number | | 7TA2 |

[a]Statistics for the highest-resolution shell are shown in parentheses.
[b]$R_{merge} = \Sigma_{hkl}\Sigma_j|I_j - <I>|/\Sigma_{hkl}\Sigma_j<I>|$, where $I_j$ and $<I>$ represent the diffraction intensity values of the individual measurements and the corresponding mean values, respectively. The summation is over all unique measurements.
[c]$R_{meas} = \Sigma_{hkl}\sqrt{(n-(n-1))}\Sigma_j|I_j - <I>|/\Sigma_{hkl}\Sigma_i<I>|$, where n is the number of diffraction intensities summated.
[d]$R_{pim} = \Sigma_{hkl}\sqrt{(1/(n-1))}\Sigma_j|I_j - <I>|/\Sigma_{hkl}\Sigma_i|<I>|$.
[e]$R_{work} = \Sigma| |F_{obs}|-|F_{calc}| | / \Sigma|F_{obs}|$, where $F_{calc}$ and $F_{obs}$ are the calculated and observed structure factor amplitudes, respectively.
$R_{free}$: statistic is the same as $R_{work}$ except calculated on [f]5.4% and [g]5.1% of the total unique reflections chosen randomly and omitted from the refinement.

Tyr34 stretches over the hollow, leaving two small cavities through which Arg37 can interact with solvent.

Ig-like β-sandwich domains are a large superfamily of proteins that share the common characteristic of two or more multi-stranded, amphipathic β-sheets interacting via a hydrophobic core[39]. The C-terminal Ig-like domain of SPACA6 follows this same pattern; it consists of two sheets (Fig. S6a). Sheet 1 is a four-stranded β-sheet (Strands D, F, H, and I) in which strands F, H, and I form an anti-parallel arrangement, and strands I and D adopt a parallel interaction. Sheet 2 is a small antiparallel two-stranded β-sheet (Strands E and G). An internal disulfide bond is observed between the C-terminal end of Strand E and the center of Strand H (Cys170-Cys226) (Fig. S6b). This disulfide bond is akin to those in β-sandwich domains from immunoglobulin proteins[40,41].

The four-stranded β-sheet twists substantially throughout its length, producing asymmetric edges that are distinct in shape and electrostatics. The thinner edge presents a flat hydrophobic surface to the environment, which stands out against the rest of the uneven and electrostatically diverse surface in SPACA6 (Fig. S6b, c). A halo of exposed backbone carbonyl/amino groups and polar side chains surrounds the hydrophobic surface (Fig. S6c). The wider edge is partially covered by a capping coiled segment that blocks the N-terminal portion of the hydrophobic core and forms three hydrogen bonds with the exposed backbone polar groups of Strand F (Fig. S6d). The C-terminal portion of this edge produces a large pocket with a partially exposed hydrophobic core. The pocket is surrounded by positive charges due to three sets of dual arginine residues (Arg162-Arg221, Arg201-Arg205, and Arg212-Arg214) and a central histidine (His220) (Fig. S6e).

**SPACA6 hinge region connects and orients the two domains**. The hinge region is a short segment between the helical and Ig-like domains that is made up of a single antiparallel three-stranded β-sheet (Strands A, B, and C), a small $3_{10}$ helix, and several long random coil segments (Fig. S7). A network of covalent and electrostatic contacts in the hinge region appear to stabilize the orientation between the 4HB and Ig-like domains. This network can be broken up into three sections. The first section involves the two CXXC motifs ($^{27}$CXXC$^{30}$ and $^{139}$CXXC$^{142}$) that form a pair of disulfides bonds between a β-hairpin in the hinge and Helix 1' in the 4HB. The second section involves an electrostatic interaction between the Ig-like domain and the hinge. Glu132 in the hinge forms salt bridges to Arg233 in the Ig-like domain and Arg135 in the hinge. The third section involves a covalent linkage between the Ig-like domain and the hinge region. Two disulfide bonds (Cys124-Cys147 and Cys128-Cys153) connect a loop in the hinge region, which is stabilized by electrostatic interactions between Gln131 and main chain functional groups, to a linker that leads into the first strand of the Ig-like domain.

**SPACA6 is structurally similar to IZUMO1**. The SPACA6 ectodomain structure and the separate 4HB and Ig-like domain structures were used to search the Protein Data Bank for structurally similar entries[42]. We identified matches with high Dali Z-scores, small root mean square deviations, and large LALI scores (the latter indicates the number of structurally equivalent residues). While the top 10 hits from the full ectodomain search (Table S1) have reasonable Z scores of >8[42], searches of only the 4HB or Ig-like domains revealed that the majority of these hits align only to the β-sandwich, a ubiquitous fold in many proteins. Only one hit was present in all three Dali searches: IZUMO1.

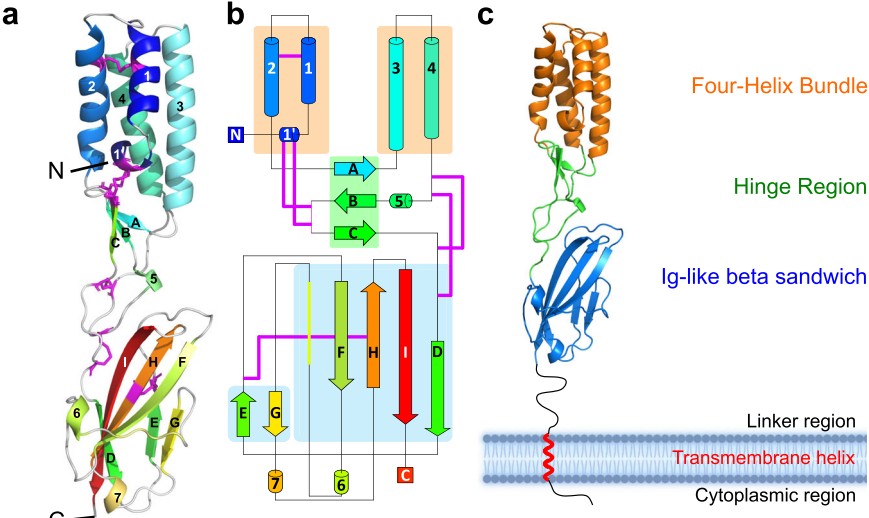

**Fig. 1 SPACA6 contains two distinct domains. a** Structure of the SPACA6 ectodomain. Ribbon diagram of the SPACA6 ectodomain, with the chain from the N to C termini colored from dark blue to dark red. Cysteines involved in disulfide bonds are colored magenta. **b** Topology diagram of SPACA6 ectodomain. The same color scheme is used as Fig. 1a. **c** Domains of the SPACA6 ectodomain. Ribbon diagram with the 4HB, hinge, and Ig-like domains colored orange, green, and blue, respectively. The membrane layer is not drawn to scale.

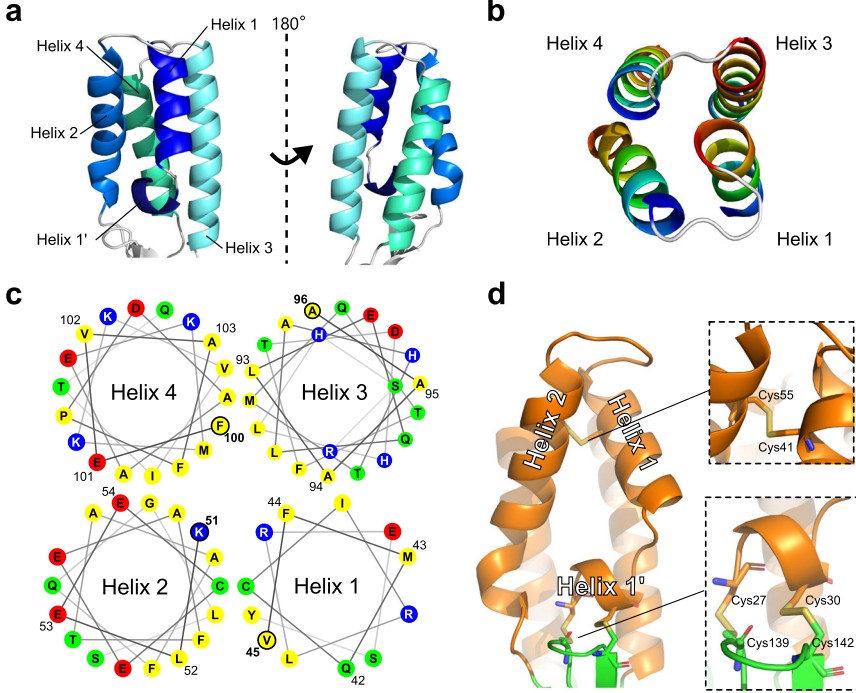

**Fig. 2 N-terminal 4HB domain of SPACA6. a** Ribbon diagram of the N-terminal 4HB. **b** Top-down view of four-helix bundle, with each helix colored dark blue at N terminus and dark red at C terminus. **c** Top-down helical wheel diagram of the 4HB with each residue presented as a circle marked with the one-letter amino-acid code; only the four amino acids that lie on the top of the wheel are numbered. Non-polar residues are colored yellow, polar non-charged residues are green, positively charged residues are blue, and negatively charged residues are red. **d** Triangular face of the 4HB domain with the 4HB colored orange and hinge colored green. The two insets display the disulfide bonds as sticks.

It has long been suspected that SPACA6 and IZUMO1 share structural similarities[7,32,37]. Despite the ectodomains of the two gamete fusion-associated proteins sharing a sequence identity of only 21% (Fig. S8a), compounding evidence including a conserved disulfide bond pattern and predicted C-terminal Ig-like domain in SPACA6 allowed for an early attempt at a homology model of mouse SPACA6, using IZUMO1 as a template[37]. Our structure confirms these predictions and reveals the true extent of the similarity. Indeed, the structure of both

SPACA6 and IZUMO1[37,43,44] share the same two-domain architecture (Fig. S8b) with similar 4HB and Ig-like β-sandwich domains connected by a hinge region (Fig. S8c).

IZUMO1 and SPACA6 4HBs share deviations from traditional helical bundles. The canonical 4HB, like those found in the SNARE protein complexes involved in endosomal fusion[45,46], has uniformly distanced helices that maintain a consistent curvature around a central axis[47]. In contrast, the coiled-coil domains in both IZUMO1 and SPACA6 are distorted with inconsistent

curvatures and uneven packing (Fig. S8d). The distortion is likely caused by the triangular shape formed by Helices 1', 1, and 2, conserved in IZUMO1 and SPACA6 and stabilized by the same CXXC motif on Helix 1'. However, an extra disulfide found in SPACA6 (the aforementioned Cys41 and Cys55 that covalently link Helices 1 and 2) produces a much sharper vertex at the top of the triangle, making SPACA6 even more distorted than IZUMO1 with a more pronounced cavity at the center of the triangle. In addition, IZUMO1 lacks the Arg37 observed in the center of this cavity in SPACA6. IZUMO1 instead has a more typical hydrophobic core of aliphatic and aromatic residues.

IZUMO1 has an Ig-like domain composed of a two-stranded and a five-stranded β-sheet[43]. The extra strand in IZUMO1 replaces the coil in SPACA6 that interacts with Strand F to cap the backbone hydrogen bonds in the strand. An interesting point of comparison is in the predicted surface charges for the Ig-like domains of these two proteins. The IZUMO1 surface is more negatively charged than that of SPACA6. The additional charges are located near the C-terminal end, which faces the sperm membrane. In SPACA6, the same areas are more neutral or positively charged (Fig. S8e). For example, both the hydrophobic surface (thinner edge) and positively charged pocked (wider edge) in SPACA6 are negatively charged in IZUMO1.

Whereas connectivities and secondary structure elements are well conserved between IZUMO1 and SPACA6, a structural alignment of the Ig-like domains revealed that the overall orientations of the two domains relative to each other are different (Fig. S9). The helical bundle of IZUMO1 is bent relative to the β-sandwich, producing a previously described "boomerang" shape that deviates by about 50° from the central axis[43]. In contrast, the helical bundle in SPACA6 has an approximately 10° lean in the opposite direction. These differences in orientation likely result from differences within the hinge region. At the primary sequence level, IZUMO1 and SPACA6 share almost no sequence similarity in the hinge save for the cysteine residues, a glycine, and an aspartate. As a result, the hydrogen-bonding and electrostatic networks are completely different. The secondary structure element of the β-sheet is shared between IZUMO1 and SPACA6, although the strands are much longer in IZUMO1, and the $3_{10}$ helix (Helix 5) is unique to SPACA6. These discrepancies result in different domain orientations of the two otherwise similar proteins.

**SPACA6 and IZUMO1 are founding members of a conserved protein superfamily.** Our Dali server search revealed that SPACA6 and IZUMO1 are the only two experimentally determined structures deposited in the Protein Data Bank that share this particular 4HB fold (Table S1). Recently, DeepMind (Alphabet/Google) developed AlphaFold, a neural network-based system that accurately predicts protein 3D structure from a primary sequence[48]. Shortly after we solved the SPACA6 structure, the AlphaFold Database was released, providing predicted structural models that cover 98.5% of all proteins in the human proteome[48,49]. Using our solved structure of SPACA6 as a search model, structural homology searching of the models in the AlphaFold human proteome identified candidates with structures potentially similar to those of SPACA6 and IZUMO1. Given AlphaFold's incredible accuracy in the prediction of SPACA6 (Fig. S10a) – especially of the ectodomain with an RMSD of 1.1 Å when compared to our solved structure (Fig. S10b) – we can have confidence that identified hits to SPACA6 are likely accurate.

Previously, PSI-BLAST searches clustered IZUMO1 with three other sperm-associated proteins: IZUMO2, IZUMO3, and IZUMO4[50]. AlphaFold predicts that these IZUMO-family proteins fold into 4HB domains with the same disulfide patterns as IZUMO1 (Figs. 3a and S11), though they lack the Ig-like domain. IZUMO2 and IZUMO3 are predicted to be single-pass membrane proteins like IZUMO1, whereas IZUMO4 appears to be secreted. Functions of IZUMO proteins 2, 3, and 4 in gamete fusion have not been established. IZUMO3 is known to play a role in the biogenesis of the acrosome during sperm development[51], and the IZUMO proteins have been observed to form complexes[50]. The conservation of IZUMO proteins in mammals, reptiles, and amphibians signals a potential function aligned with those of other known gamete fusion-associated proteins like DCST1/2, SOF1, and FIMP.

Unlike IZUMO proteins, the other SPACA proteins (i.e., SPACA1, SPACA3, SPACA4, SPACA5, and SPACA9) are predicted to be structurally divergent from SPACA6 (Fig. S12). Only SPACA9 has a 4HB, but it is not predicted to have the same parallel-antiparallel orientation as SPACA6 or the same disulfide linkages. Only SPACA1 has a similar Ig-like domain. SPACA3, SPACA4, and SPACA5 are predicted by AlphaFold to have completely different structures from SPACA6. Interestingly, SPACA4 is also known to play a role in fertilization but further upstream than SPACA6, instead aiding in the interactions between sperm and the oocyte zona pellucida[52].

Our AlphaFold searches found another match to the IZUMO1 and SPACA6 4HB, namely TMEM95. TMEM95 is a sperm-specific, single-pass transmembrane protein that when ablated leaves male mice infertile[32,33]. Sperm lacking TMEM95 have normal morphology, motility, and ability to penetrate the zona pellucida and bind the oolemma but are not able to fuse with oocyte membranes. Previous studies predicted TMEM95 to have a structural resemblance to IZUMO1[33]. Indeed, the AlphaFold model confirms TMEM95 to be a 4HB with the same pair of CXXC motifs as IZUMO1 and SPACA6, as well as the same additional disulfide between Helices 1 and 2 found in SPACA6 (Figs. 3a and S11). Whereas TMEM95 lacks an Ig-like domain, it has a region with similar disulfide bonding patterns as the hinge regions of both SPACA6 and IZUMO1 (Fig. 3b). During the publishing of this manuscript, the structure of TMEM95 was reported in a preprint server, confirming the AlphaFold results[53]. TMEM95, much like SPACA6 and IZUMO1, is evolutionary conserved as far back as amphibians (Figs. 4 and S13).

The striking overall structural similarities between SPACA6 and IZUMO1 suggests that these are the founding members of a conserved structural superfamily that includes TMEM95 and IZUMO proteins 2, 3, and 4. We propose the name *IST* superfamily after the initials of the three members known to be associated with gamete fusion so far: IZUMO1, SPACA6, and TMEM95. As only certain members possess an Ig-like domain, the hallmark feature of the IST superfamily is the 4HB domain, which has unique characteristics shared by all these proteins: 1) the distorted 4HB has helices packed in an alternating anti-parallel/parallel fashion (Fig. 5a), 2) the bundle has a triangular face made from two helices within the bundle and a third perpendicular helix (Fig. 5b), and 3) a double CXXC motif connects the perpendicular helix in the 4HB to a flexible hinge region via dual disulfide bonds (Fig. 5c). The CXXC motif, found in thioredoxin-like proteins, are known to act as redox sensors[54–56], and the motifs in IST family members might be linked to the role protein disulfide isomerases like ERp57 play in gamete fusion[57,58].

**SPACA6 ectodomain does not bind IZUMO1 or JUNO.** Given the similarities between SPACA6 and IZUMO1, the ability of the former to bind to either IZUMO1 or JUNO was tested. Biolayer interferometry (BLI) is a kinetics-based binding technique that was used previously to quantify the interaction between IZUMO1

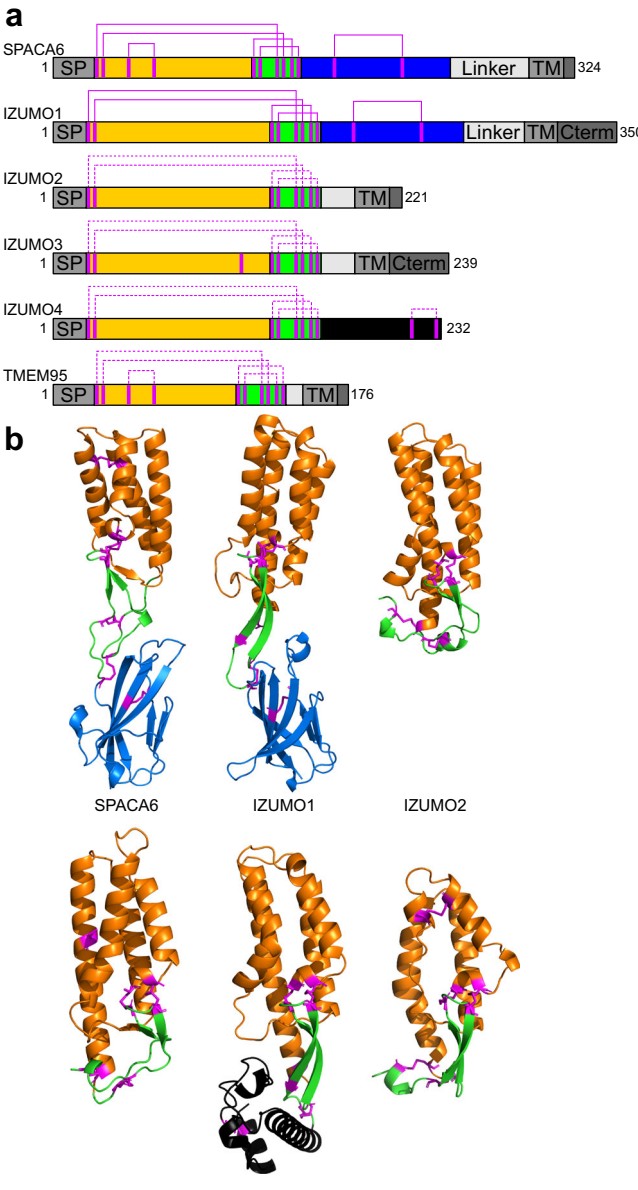

**Fig. 3 SPACA6 and IZUMO1 are founding members of a fertilization-associated superfamily. a** Domain architecture schematic of the IST superfamily with the 4HB, hinge, and Ig-like domains colored orange, green, and blue, respectively. IZUMO4 has a unique C-terminal region, which is colored black. Confirmed and putative disulfide bonds are shown in solid and dashed magenta lines, respectively. **b** Ribbon diagrams of the IST superfamily of gamete fusion-associate proteins IZUMO1 (PDB: 5F4E), SPACA6, IZUMO2 (AlphaFold DB: AF-Q6UXV1-F1), IZUMO3 (AlphaFold DB: AF-Q5VZ72-F1), IZUMO4 (AlphaFold DB: AF-Q1ZYL8-F1), and TMEM95 (AlphaFold DB: AF-Q3KNT9-F1) shown in the same color scheme as in panel A. Disulfide bonds are shown in magenta. Transmembrane helices of TMEM95, IZUMO2, and IZUMO3 are not shown.

and JUNO. Upon incubation of a biotin-labelled, sensor-bound IZUMO1 as bait with high concentrations of the JUNO analyte, a strong signal was detected (Fig. S14a), indicating a binding-induced change in the thickness of the biomaterial attached to the sensor tip. A similar signal was detected with the inverse experiment (i.e., sensor-bound JUNO as bait against IZUMO1 analyte) (Fig. S14b). No signal was detected when SPACA6 was used as the analyte against either sensor-bound IZUMO1 or sensor-bound JUNO (Fig. S14a, b). This lack of signal provides evidence that the SPACA6 ectodomain does not interact with the ectodomains of IZUMO1 or JUNO.

Since the BLI as an assay relies on biotinylation of free lysine residues on the bait protein, this modification may prevent binding if lysine residues are involved in the interaction. In addition, the binding orientation relative to the sensor may create steric hindrances; thus, traditional pull-down assays were also performed with recombinant SPACA6, IZUMO1, and JUNO ectodomains. Regardless, SPACA6 was not precipitated with either His-tagged IZUMO1 or His-tagged JUNO (Fig. S14c, d), indicating an agreement with the lack of interaction witnessed in BLI experiments. As a positive control, we confirmed interaction of JUNO with His-tagged IZUMO1 (Figs. S14e and S15).

Despite the structural similarities between SPACA6 and IZUMO1, the inability of SPACA6 to bind JUNO is not entirely surprising. There are over 20 residues on the surface of human IZUMO1 that interact with JUNO, including residues from each of the three regions (though the majority are found in the hinge region) (Fig. S14f). Of these residues, only one is conserved in SPACA6 (Glu70). While many residue substitutions maintain the original biochemical attributes, the essential Arg160 residue in IZUMO1 is changed to a negatively charged Asp148 in SPACA6; previous studies showed that an Arg160Glu mutation in IZUMO1 almost completely abolished binding to JUNO[43]. In addition, the differences in domain orientation between IZUMO1 and SPACA6 drastically augment the JUNO-binding site surface of the equivalent region on SPACA6 (Fig. S14g).

**SPACA6 surface has three patches of highly conserved residues.** Despite the known necessity of SPACA6 for gamete fusion and its similarity to IZUMO1, SPACA6 does not appear to perform the equivalent function of binding JUNO. Therefore, we sought to combine our structural data with evidence of importance provided by evolutionary biology. Sequence alignments of the SPACA6 homologs suggest a conservation of the general structure beyond mammals. For example, the cysteine residues are present even in distantly related amphibian animals (Fig. 6a). Using the ConSurf server, the conservation data from a multiple-sequence alignment of 66 sequences was mapped onto the surface of SPACA6. This type of analysis can reveal those residues that have been maintained throughout the protein's evolution and can suggest which surface areas play a role in function.

The SPACA6 structure has three highly conserved surface patches (Fig. 6b). Patch 1 spans the 4HB and the hinge region, and contains the two conserved CXXC disulfide bridges, the Arg233-Glu132-Arg135-Ser144 hinge network (Fig. S7), as well as three outward facing conserved aromatic residues (Phe31, Tyr73, Phe137). Patch 2 encompasses the wider edge of the Ig-like domain (Fig. S6e), which presents several positively charged residues toward the sperm surface. Interestingly, this patch holds an antibody epitope previously shown to prevent SPACA6 from functioning[30]. Patch 3 spans the hinge and one side of the Ig-like domain; this region has conserved prolines (Pro126, Pro127, Pro150, Pro154) and outward facing polar/charged residues. Strangely, the majority of the residues on the 4HB surface are quite variable (Fig. 6b), despite the fold's conservation throughout the SPACA6 homologs (as indicated by the bundle's hydrophobic core being conserved) and beyond into the IST superfamily.

**Conformational dynamics of SPACA6.** Although it is the smallest region of SPACA6 with the fewest definable secondary structure elements, many hinge region residues (including Patch 3) are highly conserved amongst SPACA6 homologs, perhaps indicating that the orientation of the helical bundle and

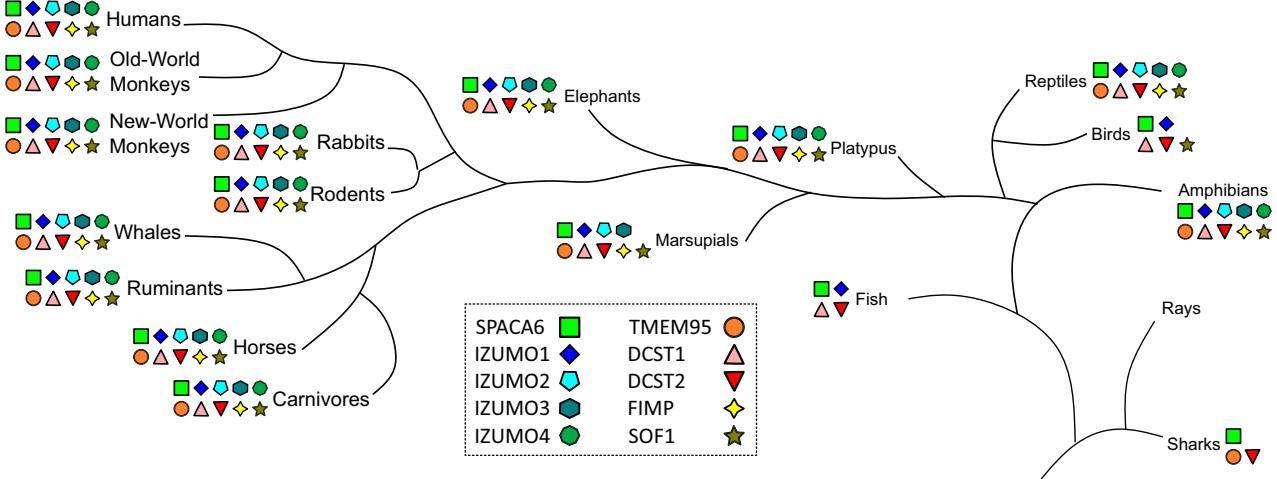

**Fig. 4 Distribution of gamete fusion-associated proteins throughout the tree of life.** PSI-BLAST searches using SPACA6, IZUMO1-4, TMEM95, DCST1, DCST2, FIMP, and SOF1 of the NCBI database were used to determine where in the tree of life these sequences are found. Distances between branch points are not drawn to scale.

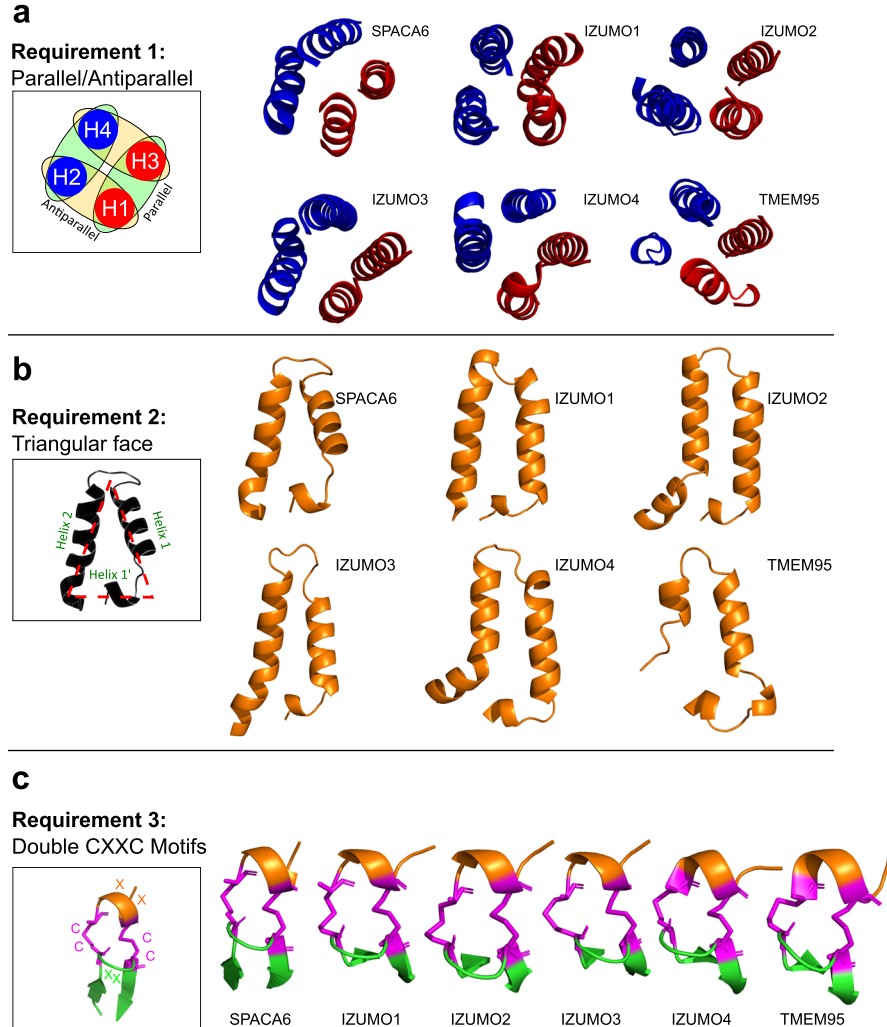

**Fig. 5 Hallmark features of the IST gamete fusion-associated protein superfamily.** Members of the IST superfamily are defined by three hallmark features of the 4HB domain: **a** four helices that alternate between parallel and antiparallel orientations, **b** a triangular face of the helical bundle, and **c** a double CXXC motif that forms two disulfides (magenta) between a small N-terminal helix (orange) and a β-hairpin (green) in the hinge region.

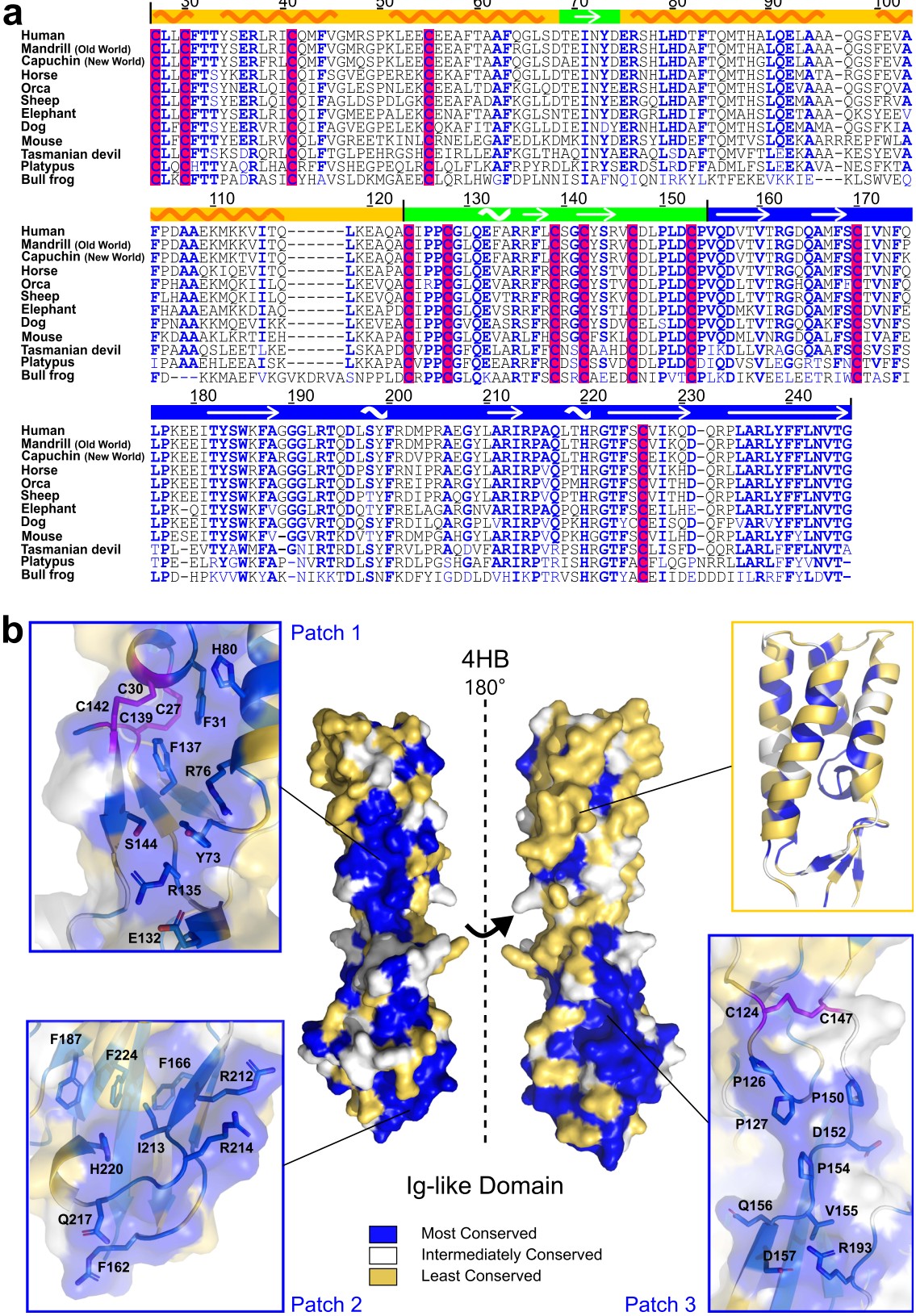

β-sandwich serves a conserved purpose. Yet, despite the extensive hydrogen bonding and electrostatic networks within the hinge regions of both SPACA6 and IZUMO1, evidence of inherent flexibility can be seen in an alignment of the multiple solved IZUMO1 structures[37,43,44]. Alignments of the individual domains overlap well, but the orientation of the domains relative to each

other varies between 50° and 70° from the central axis (Fig. S16). To understand the conformational dynamics of SPACA6 in solution, SAXS experiments were performed (Fig. S17a, b). Ab initio reconstructions of the SPACA6 ectodomain were consistent with the rod-like crystal structure (Fig. S18), though the Kratky plot reveals a level of flexibility (Fig. S17b). This conformation

**Fig. 6 Conservation of structural elements throughout SPACA6 homologs. a** Sequence alignment of SPACA6 ectodomains from twelve different species prepared using CLUSTAL OMEGA. Most conserved positions according to ConSurf analysis are colored blue. Cysteine residues are highlighted in red. Domain boundaries and secondary structure elements are shown on top of the alignment with arrows indicating beta strands and a wave indicating helices. NCBI accession IDs for included sequences are as follows: human (*Homo sapiens*, NP_001303901), mandrill (*Mandrillus leucophaeus*, XP_011821277), capuchin (*Cebus imitator*, XP_017359366), horse (*Equus caballus*, XP_023506102), orca (*Orcinus orca*, XP_012394831), sheep (*Ovis aries*, XP_014955560), elephant (*Loxodonta africana*, XP_010585293), dog (*Canis lupus familiaris*, XP_025277208), mouse (*Mus musculus*, NP_001156381), Tasmanian devil (*Sarcophilus harrisii*, XP_031819146), platypus (*Ornithorhynchus anatinus*, XP_039768188), and bull frog (*Bufo bufo*, XP_040282113). Numbering is based on the human sequence. **b** Surface representation of the SPACA6 structure oriented with the 4HB at the top and the Ig-like domain at the bottom and colored based on conservation scores from the ConSurf server. Most conserved portions are colored blue, portions with intermediate levels of conservation are white, and least conserved portions are yellow. Cysteines are colored magenta. Three surface patches showing high levels of conservation are shown in the insets labeled Patches 1, 2 and 3. A cartoon representation of the 4HB is shown in top-right inset (same color scheme).

contrasts with IZUMO1 where the unbound protein adopts a boomerang shape in the crystal lattice and in solution[43].

To specifically identify regions of flexibility, hydrogen-deuterium exchange mass spectrometry (H-DXMS) was performed on SPACA6 and compared to previously acquired data on IZUMO1[43] (Fig. 7a, b). SPACA6 is clearly more flexible than IZUMO1, as shown by the higher deuterium exchange over the entire structure after 100,000 seconds of exchange. In both structures, the C-terminal portions of the hinge region show high levels of exchange, likely allowing for limited pivoting of the 4HB and Ig-like domains relative to each other. Interestingly, the C-terminal portions of hinge section in SPACA6, comprised of residues $^{147}$CDLPLDCP$^{154}$, is the highly conserved Patch 3 (Fig. 6b), perhaps indicating that inter-domain flexibility is an evolutionary conserved trait in SPACA6. In agreement with the analysis of flexibility, CD thermal melt data showed that SPACA6 ($T_m = 51.2\,°C$) was less stable than IZUMO1 ($T_m = 62.9\,°C$) (Figs. S1e and S19).

## Discussion

The use of CRISPR/Cas9 and genetic mouse knockout strategies have led to the identification of several factors important for sperm-egg engagement and fusion[21–23,27,28,30–34]. With the exceptions of the well-characterized IZUMO1-JUNO interaction and the structure of CD9, most gamete fusion-associated proteins remain a mystery in terms of both structure and function. The biophysical and structural characterization of SPACA6 is another piece in the molecular puzzle of adhesion/fusion during fertilization.

SPACA6 and its fellow members of the IST superfamily appear to be highly conserved in mammals, as well as select birds, reptiles, and amphibians; indeed, SPACA6 is even known to be essential for fertilization in zebrafish[59]. This distribution is similar to those of other known gamete fusion-associated proteins such as DCST1[34], DCST2[34], FIMP[31], and SOF1[32], suggesting that these factors are part of a conserved molecular mechanism for fertilization used by the higher eukaryotes that lack the HAP2 (also known as GCS1) protein, which is the fusion protein responsible for catalyzing fertilization in many protists, plants, and arthropods[60,61]. Despite the strong structural similarity between SPACA6 and IZUMO1, knockouts of genes encoding either protein alone results in infertility of male mice, indicating that they are not redundant in their functions in gamete fusion[30]. More broadly, none of the known sperm proteins essential at the adhesion stage of fusion are redundant.

An open question is whether SPACA6 (as well as other members of the IST superfamily) participates in inter-gamete connections, forms intra-gamete networks that recruit essential proteins to the point of fusion, or perhaps even functions as the elusive fusogen. Co-immunoprecipitation studies in HEK293T cells suggested an interaction between full-length IZUMO1 and SPACA6[32]. However, our recombinant ectodomains failed to interact in vitro, indicating that the observed interactions in Noda et al.[32] may be mediated through their C-terminal ectodomain linker or transmembrane helix, both of which are missing in our constructs (note that the cytoplasmic tail of IZUMO1 has already been shown to be dispensable for fertilization[62]). Alternatively, IZUMO1 and/or SPACA6 may require a particular context for binding that we do not reproduce in vitro, such as a physiologically specific conformation or a molecular complex containing other proteins (known or yet to be discovered). Although the IZUMO1 ectodomain is thought to mediate an attachment of the sperm to the egg in the perivitelline space, the purpose of the SPACA6 ectodomain is unknown.

The structure of SPACA6 reveals several conserved surfaces that could participate in protein-protein interactions. The conserved portion of hinge region directly adjacent to the CXXC motifs (denoted as Patch 1 above) possesses several outward facing aromatic residues, which are commonly associated with hydrophobic and π-stacking interactions between biomolecules. The wide edge of the Ig-like domain (Patch 2) forms a positively charged trough with highly conserved Arg and His residues, and an antibody against this region was previously used to block gamete fusion[30]. The antibody recognizes the linear epitope $^{212}$RIRPAQLTHRGTFS$^{225}$, which possesses three of the six arginine residues and the highly conserved His220. Whether the function disruption resulted from occluding these specific residues or the region as a whole is unclear. The position of this cleft near the C-terminal end of the β-sandwich would suggest a *cis*-interaction with an adjacent sperm protein rather than an interaction with an oocyte protein. In addition, the conservation of the highly flexible proline-rich coil within the hinge (Patch 3) could be a protein-protein interaction site, or – perhaps more likely – indication of a conserved flexibility between the two domains important for the yet unknown role of SPACA6 in gamete fusion.

SPACA6 possesses characteristics of cell-cell adhesion proteins including the Ig-like β-sandwich. Many adhesion proteins (e.g., cadherins, integrins, adhesin, and IZUMO1) have one or more β-sandwich domains that extend the protein away from the cell membrane and toward its environmental target[63–65]. The SPACA6 Ig-like domain also contains a motif common amongst adhesion and cohesion β-sandwiches: a doublet of parallel strands at the termini of the β-sandwich known as a mechanical clamp[66]. This motif is thought to impart increased resistance to shear forces, valuable for proteins involved in interactions between cells. Yet despite these similarities to adhesins, no evidence currently exists of SPACA6 interacting with ovum proteins. The SPACA6 ectodomain is unable to bind to JUNO, as shown here, and SPACA6-expressing HEK293T cells do not form a substantial interaction with zona pellucida-free oocytes[32]. If SPACA6 does make inter-gamete connections, these interactions may require post-translational modifications or be stabilized by other sperm proteins. In support of the latter hypothesis, sperm lacking

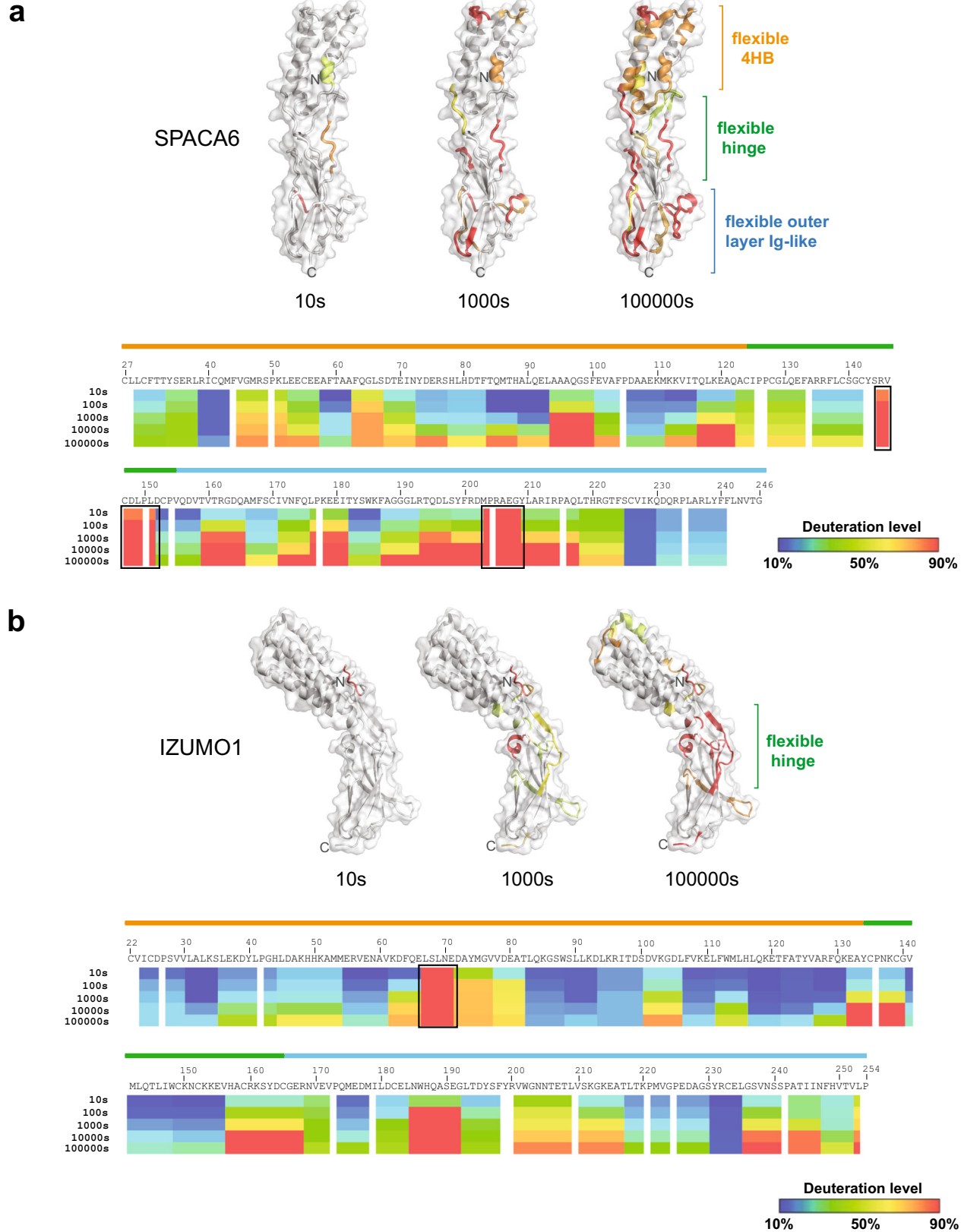

**Fig. 7 Hydrogen-deuterium exchange mass spectrometry analysis of SPACA6.** H-DXMS profiles of **a** SPACA6 and **b** IZUMO1. Percentage deuteration exchange was determined at indicated time points. Hydrogen-deuterium exchange levels are color coded in a gradient scale from blue (10%) to red (90%). Black boxes indicate areas of high exchange. The 4HB, hinge and Ig-like domain boundaries observed in the crystal structures are shown above the primary sequence. Deuterium exchange levels at 10 s, 1000 s, and 100,000 s are mapped onto a ribbon diagram overlayed with the transparent molecular surface of SPACA6 and IZUMO1. Portions of the structure with deuterium exchange levels below 50% are colored white. Areas above 50% H-DXMS exchange are colored on the gradient scale.

IZUMO1 bind to oocytes, demonstrating that molecules other than IZUMO1 are involved in gamete adhesion step[27].

Many viral, cellular, and developmental fusion proteins have characteristic features that are predictive of their function as fusogens. For example, the viral fusion glycoproteins (class I, II, and III) have hydrophobic fusion peptides or loops at the tip of the protein that insert into the host membrane. Hydropathy plots of IZUMO1[43] and structures (both solved and predicted) for the IST superfamily reveal no obvious hydrophobic fusion peptides. Thus, if any of the proteins in the IST superfamily function as fusogens, they do so in a manner distinct from other known examples.

In summary, the functions of the members of the IST super-family of gamete fusion-associated proteins remains an enticing mystery. Our characterized recombinant molecule of SPACA6 and its solved structure will provide insight into the connections between these shared structures and their roles in gamete attachment and fusion.

## Methods

**Cloning and expression of recombinant SPACA6 ectodomain.** The DNA sequence corresponding to the predicted ectodomain of human SPACA6 (NCBI Accession Number NP_001303901.1; residues 27–246) was codon optimized for expression in *Drosophila melanogaster* S2 cells and synthesized as a gene fragment (Eurofins Genomics) with an encoded Kozak sequence, BiP secretion signal, and appropriate 5′ and 3′ overhangs for ligation-independent cloning of the gene into the metallothionein promoter-based pMT expression vector modified for puromycin selection (pMT-puro). The pMT-puro vector encodes for a thrombin cleavage site followed by a C-terminal 10x-His tag (Fig. S2).

Stable transfection of the SPACA6 pMT-puro vector into *D. melanogaster* S2 cells (Gibco) was performed similar to protocols used for IZUMO1 and JUNO[43]. S2 cells were thawed and grown in Schneider's medium (Gibco) supplemented with a final concentration of 10% (v/v) heat-inactivated fetal bovine serum (Gibco) and 1X antibiotic-antimycotic (Gibco). Early passage cells were seeded ($3.0 \times 10^6$ cells) into a single well of a 6-well plate (Corning). After 24 h incubation at 27 °C, the cells were transfected with a mixture of 2 mg SPACA6 pMT-puro vector and Effectene transfection reagent (Qiagen), according to the manufacturer's protocol. Transfected cells were incubated for 72 h before the addition of 6 mg mL$^{-1}$ puromycin for selection. Subsequently, the cells were weaned off complete Schneider's medium and into serum-free Insect-XPRESS (Lonza) for large-scale protein production. Cultures of S2 cells in 1 L batches were grown to between $8$–$10 \times 10^6$ cells mL$^{-1}$ in 2 L vented flat-bottom polypropylene Erlenmeyer flasks and then induced with a final concentration of sterile-filtered 500 µM CuSO$_4$ (Millipore Sigma). Induced cultures were incubated for four days at 27 °C with shaking at 120 rpm.

**Purification of recombinant SPACA6 ectodomain.** The SPACA6-containing conditioned media was separated from cells by centrifugation at $5660 \times g$ at 4 °C prior to concentration and buffer exchange into Buffer A (10 mM Tris-HCl, pH 8.0, 300 mM NaCl, and 20 mM imidazole) using a Centramate tangential flow filtration system (Pall Corp) with a 10-kDa MWCO membrane. The concentrated SPACA6-containing media was applied onto a 2-mL column of Ni-NTA agarose resin (Qiagen). The Ni-NTA resin was washed with 10 column volumes (CV) Buffer A and then 1 CV of Buffer A supplemented with a final concentration of 50 mM imidazole. SPACA6 was eluted with 10 mL of Buffer A supplemented with a final concentration of 500 mM imidazole. Restriction-grade thrombin (Millipore Sigma) was added at 1 unit per mg of SPACA6 directly into dialysis tubing (12–14 kDa MWCO) and dialyzed against 1 L of 10 mM Tris-HCl, pH 7.5, and 150 mM NaCl (Buffer B) for 48 h at 4 °C. Thrombin-cleaved SPACA6 was then diluted three-fold to reduce the salt concentration prior to loading onto a 1 mL MonoS 5/50 GL cation-exchange column (Cytiva/GE) equilibrated with 10 mM Tris-HCl, pH 7.5. The cation exchanger was washed with 3 CV 10 mM Tris-HCl, pH 7.5 prior to eluting SPACA6 using a linear gradient of 0 to 500 mM NaCl in 10 mM Tris-HCl, pH 7.5 over 25 CV. Following ion-exchange chromatography, SPACA6 was concentrated to 1 mL and isocratically eluted from an ENrich SEC650 10 × 300 column (BioRad) equilibrated with Buffer B. Based on the chromatogram, fractions containing SPACA6 were pooled and concentrated. Purity was monitored via Coomassie-stained 16% SDS-polyacrylamide gel electrophoresis. Protein concentration was quantified based on absorbance at 280 nm using the Beer-Lambert Law and theoretical molar extinction coefficient.

**Circular dichroism spectroscopy.** Purified SPACA6 was dialyzed overnight into 10 mM sodium phosphate, pH 7.4, and 150 mM NaF and diluted to 0.16 mg mL$^{-1}$ prior to analysis by CD spectroscopy. CD wavelength spectral scans were collected at 25 °C between 185 to 260 nm at a rate of 50 nm/min using a 1 mm pathlength

quartz cuvette (Helma) in a Jasco J-1500 spectropolarimeter. CD spectra were baseline corrected, averaged over 10 accumulations, and converted to mean residue ellipticity ($\theta_{MRE}$) in units of degree cm2·dmol−1:

$$\theta_{MRE} = \frac{MW}{(N-1)} \cdot \frac{\theta}{(10.d.c)}$$

where $MW$ is the molecular weight for each sample in Da; $N$ is the number of amino acids; $\theta$ is the ellipticity in millidegree; $d$ corresponds to the optical path-length in cm; and $c$ is the protein concentration in mg mL$^{-1}$.

Thermal denaturation assays were performed at a wavelength of 207 nm by increasing the temperature from 20 to 80 °C in 5 °C intervals with 2 min equilibration between temperature points. Four scans were taken per temperature point, averaged, and baseline corrected. The resultant change in ellipticity was normalized between 0 (folded) and 1 (unfolded) and fit to a non-linear biphasic sigmoidal curve using the program GraphPad (version 8.4.3) to determine the apparent melting temperature ($T_m$). Secondary structural composition (helices, strands, turns and coils) was estimated using the CONTIN method and the Set3 library (185–240 nm) using the Dichroweb server[67].

**SEC-MALS analysis.** The oligomeric state of tag-removed, fully glycosylated SPACA6 was assessed by SEC-MALS. 0.14 mg Bovine Serum Albumin (BSA) and 0.14 mg SPACA6 were prepared in 1X PBS at a concentration of 1.2 mg mL$^{-1}$. A Superdex 75 10/300 GL size-exclusion column (Cytiva/GE) was equilibrated overnight with 5 CV PBS. Monomeric BSA (MW = 66,432 Da) was used as a reference calibration standard. Prior to SEC-MALS analysis, each sample was centrifuged at $15,000 \times g$ for 15 min at 4 °C and then the supernatant was loaded onto the size-exclusion column on an AKTA Pure FPLC (Cytiva) at 0.2 mL min$^{-1}$. Triple detection was performed by measuring absorbance at 280 nm using the integrated UV monitor on the AKTA Pure, three-angle light scattering using the miniDAWN TREOS MALS detector (Wyatt) and refractive index (RI) using Optilab T-rEX RI detector (Wyatt). The data were processed, and weight-averaged molecular mass was calculated using the ASTRA software package (version 7.0.2.11).

**Sparse matrix crystallization and data collection.** Sitting-drop sparse matrix crystallization screening was performed using a Douglas Instrument Oryx8 liquid handling system. SPACA6 concentrated to 7 mg mL$^{-1}$ in Buffer B was mixed in a 1:1 volumetric ratio (0.3:0.3 mL) with JCSG + (Qiagen), Cryos (Qiagen), and PACT (Qiagen) sparse matrix crystallization screens in 96-well 2-drop Art Robbins Intelliplates. Protein crystal clusters were obtained when SPACA6 was mixed with 100 mM Bis-Tris, pH 5.5, 200 mM NaCl, 25% (w/v) PEG 3350. To improve crystal morphology, the crystal clusters were crushed into microcrystals using the Seed Bead kit (Hampton Research) and used for random microseed matrix screening[38]. SPACA6 (7 mg mL$^{-1}$), microseeds, and sparse matrix crystallization conditions from the JCSG+, and Cryos suites were mixed in a 3:1:2 ratio. From this procedure, larger clusters were consistently obtained in 200 mM ammonium chloride and 20% (w/v) PEG 3350. As initial X-ray diffraction was poor, a detergent screen (Hampton Research) was used to identify conditions that yielded strongly diffracting crystals. The crystals used to solve the structure were obtained by the microseeding protocol using a precipitant solution composed of 200 mM ammonium chloride, 20% (w/v) PEG 3350, and 34 mM FOS-choline-8. Crystals were looped and coated with perfluoropolyether cryo oil (Hampton Research) before being flash cooled in liquid nitrogen (100 K). Multiple native datasets were collected at the AMX (17ID-1) beamline at the National Synchrotron Light Source-II (NSLS-II) at Brookhaven National Laboratory (Upton, NY). Data were indexed and integrated using DIALS[68] and averaged and scaled using Aimless in the CCP4 suite[69].

**Structure determination.** No usable molecular replacement model was available, be it solved structure of AlphaFold model. Therefore, attempts to solve the structure of SPACA6 via SAD phasing were made, with multiple datasets collected at the NE-CAT (24-ID-C) beamline at the Advanced Photon Source (Argonne National Laboratory, Lemont, IL); phasing attempts with tantalum clusters and/or intrinsic sulfurs were fruitless. Eventually, initial phases were determined using the human IZUMO1 model (PDB: 5JK9) as a starting template for ARCIMBOLDO_SHREDDER[70,71]. ARCIM-BOLDO_SHREDDER carried out expected log-likelihood gain-guided molecular replacement of template-generated fragments using the program Phaser[72]. During molecular replacement, additional refinement, such as gyre refinement against the rotation function, and gimble refinement after translation, was employed to refine the internal geometry of the fragments, allowing additional degrees of freedom by sub-division in rigid groups[73]. Consistently placed and refined fragments were combined in reciprocal space with ALIXE[74]. Best-scored phase sets were subject to density modification and autotracing in SHELXE[75]. The solution from ARCIMBOLDO_SHREDDER provided clear density for ~65% of the structure, successfully built with a *CC* of 41.6%. X-ray data collection statistics are presented in Table 1.

To complete the model, Br-SAD phasing was performed and combined with the phases from the ARCIMBOLDO_SHREDDER model. SPACA6 crystals were soaked in 200 mM ammonium chloride, 20% (w/v) PEG 3350, 34 mM FOS-choline-8, 25% (v/v) glycerol, and 1 M sodium bromide for 10 s before flash cooling

directly into liquid nitrogen. Data were collected on the NE-CAT (24-ID-C) beamline at the Advanced Photon Source (Argonne National Laboratory, Lemont, IL). Data were indexed and integrated using DIALS[68], followed by scaling and merging using Aimless from the CCP4 suite[69]. X-ray data collection statistics are presented in Table 1. The Br-SAD dataset is not isomorphous to the native dataset, likely due to cryo solution-induced dehydration of the crystal. PHENIX_Autosol was used to carry out a combination of molecular replacement and SAD phasing[76,77]. The atomic positions of ten bromide ions were found using a hybrid substructure search in PHENIX_Autosol. Model phases calculated from a molecular replacement solution using the ARCIMBOLDO model was combined with the anomalous signal from the Br-SAD data. PHENIX_AutoBuild was used to build the majority (~80%) of the SPACA6 model. The final PHENIX_AutoBuild model was subjected to an initial round of simulated annealing torsion angle refinement starting at 5000 K using PHENIX_refine[78]. The complete SPACA6 model was built with iterative rounds of manual model building followed by coordinate refinement using Coot[79] and PHENIX_refine[78], respectively. Riding hydrogens and Translation/Liberation/Screw groupings were included to improve the structure. The program PDB-REDO[80] was used to provide a final geometry clean up prior to model validation using Ramachandran, geometry, and rotamer analysis, difference map peak searches in Coot[79], and MolProbity[81] via PHENIX. All structural images were generated using The PyMOL Molecular Graphics System (Version 2.0 Schrödinger, LLC). Electrostatic calculations were done using the APBS web server[82].

**Biolayer interferometry**. The binding affinities of SPACA6 to IZUMO1 and JUNO were measured by BLI using a single-channel BLItz instrument (FortéBio/Sartorius). Purified JUNO and IZUMO1 in PBS were biotinylated using the EZ link sulfo-NHS-LC-biotinylation kit (Thermo Pierce), according to the manufacturer's instructions, though altered to use a 1:1 protein to biotin ratio. Excess biotin reagent was removed using a 5 mL HiTrap Desalting column (Cytiva) equilibrated in PBS. All streptavidin-coated biosensors were hydrated in BLI rehydration buffer (PBS, 0.5 mg mL$^{-1}$ BSA, and 0.01% (v/v) Tween-20) for 10 min. Biotinylated bait proteins were diluted in BLI kinetics buffer (PBS, 0.1 mg mL$^{-1}$ BSA, and 0.01% (v/v) Tween-20) to a final concentration of 20 µg mL$^{-1}$ and immobilized onto a streptavidin-coated biosensor for 90 s. Analyte proteins were prepared in BLI kinetics buffer at high concentrations (IZUMO1, 12.5 µg mL$^{-1}$; JUNO, 12.5 µg mL$^{-1}$; SPACA6, 150 µg mL$^{-1}$); analyte association to bait was measured over 120 s at 20 °C. Subsequently, the streptavidin-coated biosensor was immersed into BLI kinetics buffer for 120 s to dissociate the analyte. BSA and BLI kinetics buffer only against streptavidin-coated biosensors loaded with biotinylated bait were used to quantify non-specific binding. The data were analysed and sensorgrams were step corrected, reference corrected, and fit to a 1:1 binding model. Reported results are representative of independent duplicate trials.

**Pull down assay**. Bait protein (IZUMO1 or JUNO) tagged with 10x-His tags were mixed with target protein (JUNO or SPACA6) in a 1:1 molar ratio in Buffer B at a total volume of 100 µL. The samples were incubated on a rotator for 1 h at room temperature and then incubated with 15 µL of a 25% slurry of nickel-charged MagBeads (Genscript) for an additional 1 h on the rotator at room temperature. Magnetic beads were captured using the PureProteome magnetic stand (Millipore Sigma), and the supernatant was removed. The MagBeads were then washed five times with 100 µL of Buffer B plus 25 mM imidazole. Bound protein was then eluted with 100 µL of Buffer B plus 400 mM imidazole and separated on 20% polyacrylamide gels for SDS-PAGE analysis. Protein bands were detected by Coomassie Brilliant Blue staining and anti-His western blot. For the western blot, protein was transferred from the 20% polyacrylamide gels to PVDF Immobilon-P membranes (Millipore Sigma) and blocked with 5% (w/v) skim milk in PBS with 0.1% (v/v) Tween 20 (PBST). Transfer membranes were then incubated with a 1:10,000 dilution of primary mouse anti-6xHis antibody (Roche, cat. #11922416001) for 1 h, followed by 1:10,000 secondary HRP-conjugated goat anti-mouse IgG (H + L) antibody (Invitrogen, cat. #62-6520) for 1 h with three 10 min PBST washes in between. The membranes were developed using UltraScence Pico Western Substrate (BIO-HELIX), and its chemiluminescence signal was imaged using a G:Box gel documentation system (Syngene). Anti-His primary was validated by manufacturer's (Roche), with the lot number used confirmed as "meeting our specification" in the Certificate of Analysis. Reported results are representative from independent duplicate trials.

**Small angle X-ray scattering**. SEC-SAXS was performed at NSLS-II using their mail-in service on the Life Sciences X-ray Scattering 16-ID beamline[83–85]. The SPACA6 ectodomain was dialyzed into commercially prepared 1X PBS (Millipore Sigma) and concentrated to ~6.5 mg mL$^{-1}$ prior to shipment on ice. The protein was centrifuged at 20,000 × $g$ for 10 min before loading 45 µL onto a Superdex 200 Increase 5/150 GL column (Cytiva) at 0.5 mL min$^{-1}$ on a Shimadzu bio-inert HPLC system for in-line SAXS measurements. Flow from the column was split 2:1 using a passive splitter between the X-ray scattering measurements and the UV/Vis and refractive index detectors. Subsequent buffer subtraction, peak selection, and profile analysis was undertaken using Lixtools[83] and the ATSAS (3.0.3) software package[86]. Ab initio reconstructions of SPACA6 were performed using DAMMIN[87,88]. Twenty independent DAMMIN models were superimposed and averaged using DAMAVER[89] to obtain a consensus molecular envelope. The final SPACA6 structure was superimposed onto the SAXS molecular envelope using SUPALM function as part of the SASpy PyMOL plugin[90].

**Hydrogen/deuterium-exchange mass spectrometry**. Deuterium exchange was initiated by mixing 25 µL of untagged SPACA6 ectodomain (4 mg mL$^{-1}$) with 75 µL D$_2$O buffer (8.3 mM Tris-HCl, pH 7.2, 150 mM NaCl in D$_2$O, pD$_{read}$ 7.2). Spectra were obtained after incubation at 0 °C for 10, 100, 1000, 10,000, and 100,000 s. At these times, 16 µL of exchange samples were added to 24 µL quench solution to stop the D$_2$O exchange reaction. After 5 min incubation on ice, quenched samples were diluted by addition of 129 µL of ice-cold dilution buffer, and then immediately frozen on dry ice and stored at −80 °C. The non-deuterated control samples and equilibrium-deuterated control samples were also prepared by mixing protein with 8.3 mM Tris-HCl, pH 7.2, 150 mM NaCl in H$_2$O or with 0.8% (v/v) formic acid in 99.9% D$_2$O. The frozen samples were then thawed at 5 °C and passed over an immobilized pepsin column (16 µL bed volume) at a flow rate of 25 µL min$^{-1}$. The resulting peptides were collected on a C$_{18}$ trap for desalting and separated by a nanoEase M/Z Peptide BEH C18 reverse phase column (Waters, 0.3 × 50 mm, 130 A, 17 µm) using a linear gradient of acetonitrile from 5% to 45% over 30 min. MS analysis was performed using the OrbiTrap Elite Mass Spectrometer (Thermo Scientific) with a capillary temperature of 200 °C. Data were acquired in both data-dependent MS/MS mode and MS1 profile mode, and the data were analysed by Proteome Discoverer software and HDExaminer (Sierra Analytics Inc.).

**Statistics and reproducibility**. For CD thermal melts, the results are presented as mean±standard deviation (SD). Independent triplicates, $n = 3$, were performed for all CD spectroscopy studies. Independent duplicates, $n = 2$, were performed for BLI and pull downs studies.

**Reporting summary**. Further information on research design is available in the Nature Research Reporting Summary linked to this article.

## Data availability
Further information and requests for resources and reagents are available from the corresponding author on reasonable requests. Atomic coordinates and structure factors for SPACA6 have been deposited in the Protein Data Bank (PDB) with the accession code: 7TA2. SAXS data have been deposited in the Small Angle Scattering Biological Data Bank (SASBDB) with the accession code: SASDNM3. Source data are provided with this paper, including unedited/uncropped gels in Fig. S15.

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

## Acknowledgements

We thank Dr. Vitor Hugo B. Serrão and Dr. Karen K. Siu for their assistance with the biophysical studies and the crystallography, respectively. We thank the staff on Beamline 16ID (LiX) and 17ID-1 (AMX) at the National Synchrotron Light Source II (NSLS-II), and Beamline 24ID-C (NE-CAT) at the Advanced Photon Source (APS) for synchrotron access and support. This work was supported by funding from the Canadian Institutes of Health Research (CIHR; PJT-153281) and Canada Research Chair (CRC-2017-00140) to J.E.L. and by a joint grant to J.E.L. and A.Z. from the New Frontiers in Research Fund (NFRFE-2019-00230). Biophysics and structural biology infrastructure were supported by funding from the Canada Foundation for Innovation-John R. Evans Leaders Fund. T.D.R.V. was supported by a CIHR Postdoctoral Fellowship. The phasing of the structure is based upon research conducted at the Northeastern Collaborative Access Team beamlines, which are funded by the National Institute of General Medical Sciences from the National Institutes of Health (P30 GM124165). This research used resources of the Advanced Photon Source, a U.S. Department of Energy (DOE) Office of Science User Facility operated for the DOE Office of Science by Argonne National Laboratory under Contract No. DE-AC02-06CH11357. Support for work performed at the Center for Biomolecular Structure beamline LIX (16ID) | AMX (17ID-1) | FMX (17ID-2) at NSLS-II is provided by NIGMS-1P30GM133893 and BER-BO 070. NSLS-II is supported by DOE, BES-FWP-PS001.

## Author contributions

J.E.L. and A.Z. conceived the project; T.D.R.V. and J.E.L. designed the project, discussed all results, and wrote the manuscript; T.D.R.V. collected the X-ray diffraction data, completed and refined the crystal structure, performed biochemical characterization, determined the SAXS reconstructions, and analyzed the structure; T.D.R.V. and P.Y. performed the expression, purification, and crystallization experiments; I.U. and E.J. obtained the initial ARCIMBOLDO SPACA6 model; S.L. performed the H/DXMS experiments; D.G. performed the pull-down experiments; J.B. collected the SEC-SAXS data; all authors edited and approve the manuscript.

## Competing interests

The authors declare no competing interests.

## Additional information

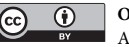

