## [Peer Review File · Communications Biology]

Reviewers' comments:

Reviewer #1 (Remarks to the Author):

The submitted work of Vance et al. provides clear structural details of SPACA6 which is a sperm-expressed surface protein and considered critical for gamete fusion during mammal sexual reproduction. The authors have used different biophysical techniques along with bioinformatics analysis to shed key insights on SPACA6 structure in addition to its high resolution X-ray structure. They have used Alpha fold prediction followed by a display of conservation of structural elements throughout SPACA6 homologs to propose a new IST superfamily for gamete fusion-associated proteins. Answering following comments might help authors improve their manuscript.

1. Authors should add IZUMO1 as founding member of IST superfamily in the abstract similar to their manuscript's introduction.
2. Authors could not see interaction of SPACA6 with IZUMO1 or JUNO. Authors have tried to explain this contrasting observation in comparison to cited reference 31 (Noda et. al. PNAS, 2020). Authors line of argument considers structural similarity of SPACA6 and IZUMO1. But authors themselves show significant differences in structures of SPACA6 and IZUMO1 in addition to differences in their conformational dynamics by HDX-MS (Figure 7). Could these differences explain lack of binding between SPACA6 and IZUMO1/JUNO?
3. Continuing to previous point, authors suggest their recombinant SPACA6 ectodomain failed to interact with IZUMO1/JUNO and interaction requires transmembrane helices or linkers. Could author compare more clearly their purified SPACA6 ectodomain with that of Noda et. al. This will help readers understand differences in authors and literature observations.
4. In addition to points 2 and 3, could there be yet undiscovered interaction partners of SPACA6 and possibly SPACA6 does not bind to JUNO in vitro too at all. Authors could consider repeating co-immunoprecipitation studies in HEK293T cells with their purified SPACA6 ectodomain. If it's not possible, authors could consider discussing above with their results.
5. Authors show Alphafold predicted structure of different proteins involved in proposed gamete fusion associated superfamily. It will be interesting to learn authors comments on the role of these members and their presence across evolution (Figures 3 & 4).
6. Authors mention different approaches for SOACA6 expression, crystallization and solving X-ray structure. They have previously published the X-ray structure of human sperm IZUMO1 and egg JUNO fertilization complex in reference 40 (Aydin et al, Nature 2016). It will be helpful to know whether challenges faced in solving SPACA6 structure were similar to IZUMO1 structure. Should interested readers follow some guidelines for structural studies of IST superfamily members?

Reviewer #2 (Remarks to the Author):

In this manuscript, Vance et al. solve the crystal structure of an essential factor in vertebrate fertilization: SPACA6. Through structural similarity searches, they confirm the presence of conserved structural motifs present in the Izumo family of proteins, Spaca6 and Tmem95, all of which play a crucial role in gamete binding and fusion. While these similarities had already been proposed and described in previous publications (Nishimura et al, 2016; Lamas-Tomanzo, 2020), this work provides a confirmation of these claims and performs a thorough structural analysis. This work represents important groundwork that is necessary to elucidate the role of these proteins in the process of fertilization.

The manuscript can be improved by addressing the following concerns:

1. Given the recent development of AlphaFold, the authors should comment and show overlays on how the crystal structure of SPACA6 compares to the prediction by AlphaFold2.
2. The authors describe a new superfamily of proteins which they term IST. These proteins have been predicted to be structurally similar in previous publications (Nishimura et al, 2016; Lamas-Tomanzo, 2020). To provide an accurate historical background of the field, the authors should phrase their findings in the context of these publications and describe how their work provides confirmation of these predictions and associations. As currently written, the authors underscore the previous findings in the field. Specifically, the following claims should be adjusted: lines 80-82, lines 177-183
3. Some claims in the manuscript are missing references or do not provide comprehensive background information in the field:
 - a. In line 54, when describing the role of CD9 in fertilization, please reference work by Inoue et al. describing CD9's function in membrane compartmentalization. doi: 10.1242/dev.189985.
 - b. In line 63, include work by Noda et al. on DCST1/2. doi: 10.1038/s42003-022-03289-w.
4. In addition to biolayer interferometry, the authors perform pulldowns to assess whether SPACA6 and IZUMO1 interact through their ectodomain and show that SPACA6 was not precipitated with recombinant IZUMO1. An important control to add is a negative control to show that the His tag does not lead to non-specific interactions.
5. In line 7, the claim that all proteins in the IST superfamily are involved in gamete fusion is not accurate. The other members of the IZUMO family have not been shown to be essential for gamete fusion.
6. The Discussion could benefit from an in-depth analysis of the functional sites of SPACA6 and their potential functional roles.

Reviewer #3 (Remarks to the Author):

SPACA6 is a recent identified sperm protein that involves in sperm-egg engagement and fusion. In this study, the authors report the high resolution crystal structure of the ectodomain of SPACA6 and find that it is highly similar to IZUMO1. They thus identified a superfamily of proteins, including IZUMO1, SPACA6, and TMEM95, that have common 4HB domain structure feature. Although structurally similar to IZUMO1, SPACA6 cannot bind to JUNO. Overall, the data is clean and solid. Although the structure does not yet answer how SPACA6 mediates the sperm-egg fusion, it provides an important framework to understand its structural features that may be shared among other gamete fusion-associated proteins.

Below are some specific comments:

1. The author showed that SPACA6 cannot interact with JUNO. It is better to provide a structural explanation for such phenomenon. For example, are the interface or key residues that mediate the interaction between IZUMO1 and JUNO conserved in SPACA6?
2. It is pity that the potential SPACA6 interacting ovum protein remains unknown. Since the structure of SPACA6 is similar to IZUMO1, it is not surprising that the SPACA6 interacting protein may be structurally similar to JUNO. Have the authors also performed a DALI search for JUNO? It would be nice to have more discussion on such perspectives in searching for the binding partners of SPACA6.
3. It is pleased to see that the authors further studied the conformational dynamic of SPACA6 using SAXS and H-DXMS methods. One wonder whether such conformational dynamic has any functional relevance?
4. Is the recombinant expressed SPACA6 glycosylated? Does the author observe the glycosylation site

in the structure?

Minors:

1. The manuscript contains too many figures, some of which can be obviously combined (e.g. S14 and S15).
2. The structural difference seen in S10 between SPACA6 and IZUMO1 cannot be recognized as structural flexibility as they are from two different targets.

Point-by-Point Response

Reviewer #1:

1. Authors should add IZUMO1 as founding member of IST superfamily in the abstract similar to their manuscript's introduction.

Response: As requested, IZUMO1 – indeed a founding member of the IST superfamily – has been added to the abstract as follows (Lines 6-8): **“This structure is reminiscent of IZUMO1, another gamete fusion-associated protein, making SPACA6 and IZUMO1 founding members of a superfamily of fertilization-associated proteins, herein dubbed the IST superfamily.”**

2. Authors could not see interaction of SPACA6 with IZUMO1 or JUNO. Authors have tried to explain this contrasting observation in comparison to cited reference 31 (Noda et. al. PNAS, 2020). Authors line of argument considers structural similarity of SPACA6 and IZUMO1. But authors themselves show significant differences in structures of SPACA6 and IZUMO1 in addition to differences in their conformational dynamics by HDX-MS (Figure 7). Could these differences explain lack of binding between SPACA6 and IZUMO1/JUNO?

Response: We thank the reviewer for their suggestions on how to explain the discrepancies between our *in vitro* work and the work of Noda *et al.* Indeed, the lack of binding between SPACA6 and IZUMO1 ectodomains in our hands stands in contrast to that of previously published Co-IP data. While the structural differences between SPACA6 and IZUMO1 should be present in both our ectodomains and the full-length constructs used by Noda *et al.*, differences in physiological conformation or dynamics could explain this difference, as stated in Line 377-379: “Alternatively, IZUMO1 and/or SPACA6 may require a particular context for binding that we do not reproduce *in vitro*, such as a physiologically specific conformation or a molecular complex containing other proteins (known or yet to be discovered).”

The inability of SPACA6 to bind JUNO on the other hand is not unexpected, as even Noda *et al.* were unable to note significant interactions between SPACA6-expressing HEK293 cells and ZP-free oocytes. A structural reason for this inability to bind is now provided in the text (Line 288-296) and in **Figure S14**.

3. Continuing to previous point, authors suggest their recombinant SPACA6 ectodomain failed to interact with IZUMO1/JUNO and interaction requires transmembrane helices or linkers. Could author compare more clearly their purified SPACA6 ectodomain with that of Noda et. al. This will help readers understand differences in authors and literature observations.

Response: In an effort to clarify this point, we have augmented the following sentences (Lines 371-376): “Co-immunoprecipitation studies in HEK293T cells suggested an interaction between full-length IZUMO1 and SPACA6³². However, our recombinant ectodomains failed to interact *in vitro*, indicating that the observed interactions in Noda *et al.* may be mediated through their C-terminal ectodomain linker or transmembrane helix, both of which are missing in our constructs (note that the cytoplasmic tail of IZUMO1 has already been shown to be dispensable for fertilization⁶²).”

4. In addition to points 2 and 3, could there be yet undiscovered interaction partners of SPACA6 and possibly SPACA6 does not bind to JUNO in vitro too at all. Authors could consider repeating co-immunoprecipitation studies in HEK293T cells with their purified SPACA6 ectodomain. If it's not possible, authors could consider discussing above with their results.

Response: We agree with the reviewer's proposition that it is very likely that SPACA6 does not bind JUNO. The structural reasons for this are shown in the added panels F and G of Figure S14. Identifying the interacting partner for SPACA6 is an interest of our lab. While the use of HEK293T cells are commonly used for co-IP and discovery of protein-protein interactions, this cell type is not ideal in the discovery of proteins involved in fertilization. Many fertilization factors are exclusively expressed on gametes and its expression is tightly regulated. The most appropriate experimental design would be to use human oocytes for the protein-protein interaction studies. However, obtaining enough healthy human oocytes for research is extremely difficult due to scarcity of supply, restrictions of use, among other factors. Thus, we believe the co-IP experiments with our recombinant SPACA6 is beyond the scope of this manuscript.

Instead, we have changed our final conclusions to read (Lines 420-423): "In summary, the functions of the members of the IST superfamily of gamete fusion-associated proteins remains an enticing mystery. Our characterized recombinant molecule of SPACA6 and its solved structure will provide insight into the connections between these shared structures and their roles in gamete attachment and fusion."

5. Authors show Alphafold predicted structure of different proteins involved in proposed gamete fusion associated superfamily. It will be interesting to learn authors comments on the role of these members and their presence across evolution (Figures 3 & 4).

Response: We share the reviewer's interest in answering the question of what the IST superfamily members' roles are, especially given their conservation across a huge range of organisms. However, given the paucity of information regarding their functional roles, we prefer to minimize our speculation, and keep our Discussion more general (Line 360-363): ". . . suggesting that these factors are part of a conserved molecular mechanism for fertilization used by the higher eukaryotes that lack the HAP2 (also known as GCS1) protein, which is the fusion protein responsible for catalyzing fertilization in many protists, plants, and arthropods^{60,61}."

6. Authors mention different approaches for SOACA6 expression, crystallization and solving X-ray structure. They have previously published the X-ray structure of human sperm IZUMO1 and egg JUNO fertilization complex in reference 40 (Aydin et al, Nature 2016). It will be helpful to know whether challenges faced in solving SPACA6 structure were similar to IZUMO1 structure. Should interested readers follow some guidelines for structural studies of IST superfamily members?

Response: We agree that a set of guidelines for structural studies of IST superfamily members would be ideal. However, the required strategies for crystallization and structure solution of SPACA6 were quite different from those used for the solution of IZUMO1. While the structure determination of IZUMO1 was also difficult, molecular replacement did find a correct solution for the Ig-like domain. This provided enough phasing power to provide an interpretable electron density map to build the entire IZUMO1 structure. For SPACA6, extensive molecular replacement studies involving an exhaustive list of complete and truncated Ig-like and 4HB domains failed to identify a solution. Sadly, we believe each IST superfamily member will likely require its own ground-up approach to structure solution.

Reviewer #2:

1. Given the recent development of AlphaFold, the authors should comment and show overlays on how the crystal structure of SPACA6 compares to the prediction by AlphaFold2.

Response: We thank the reviewer for their suggestion and have added a supplementary figure comparing SPACA6's solved structure to the AlphaFold prediction (Figure S10). We have also added the sentence (Lines 218-220): "Given AlphaFold's incredible accuracy in the prediction of SPACA6 (Figure S10A) – especially of the ectodomain with an RMSD of 1.1 Å when compared to our solved structure (Figure S10B) – we can have confidence that identified hits to SPACA6 are likely accurate."

2. The authors describe a new superfamily of proteins which they term IST. These proteins have been predicted to be structurally similar in previous publications (Nishimura et al, 2016; Lamas-Tomanzo, 2020). To provide an accurate historical background of the field, the authors should phrase their findings in the context of these publications and describe how their work provides confirmation of these predictions and associations. As currently written, the authors underscore the previous findings in the field. Specifically, the following claims should be adjusted: lines 80-82, lines 177-183.

Response: We have taken the reviewer's comments to heart and have augmented our manuscript to provide better context for the previous studies that predicted the similarities between IZUMO1 and SPACA6. We made multiple changes throughout the text:

Line 67: add the sentence "**As predicted in previous studies** ^{7,32,37}, the domain architecture of SPACA6 is similar to that of human IZUMO1 . . ."

Line 166-173: reorganized and added to the paragraph "It has long been suspected that SPACA6 and IZUMO1 share structural similarities ^{7,32,37}. **Despite the ectodomains of the two gamete fusion-associated proteins sharing a sequence identity of only 21% (Figure S8A), compounding evidence including a conserved disulfide bond pattern and predicted C-terminal Ig-like domain in SPACA6 allowed for an early attempt at a homology model of mouse SPACA6, using IZUMO1 as a template** ³⁷. **Our structure confirms these predictions** and reveals the true extent of the similarity. Indeed, the structure of both SPACA6 and IZUMO1 ^{37,43,44} share the same two-domain architecture (Figure S8B) with similar 4HB and Ig-like β -sandwich domains connected by a hinge region (Figure S8C)."

Line 241-248: reorganized and added to the paragraph: "Our AlphaFold searches found another match to the IZUMO1 and SPACA6 4HB, namely TMEM95. TMEM95 is a sperm-specific, single-pass transmembrane protein that when ablated leaves male mice infertile ^{32,33}. Sperm lacking TMEM95 have normal morphology, motility, and ability to penetrate the zona pellucida and bind the oolemma but are not able to fuse with oocyte membranes. **Previous studies predicted TMEM95 to have a structural resemblance to IZUMO1** ³³. Indeed, the AlphaFold model confirms TMEM95 to be a 4HB with the same pair of CXXC motifs as IZUMO1 and SPACA6, as well as the same additional disulfide between Helices 1 and 2 found in SPACA6 (Figure 3A and Figure S11)."

3. Some claims in the manuscript are missing references or do not provide comprehensive background information in the field:

a. In line 54, when describing the role of CD9 in fertilization, please reference work by Inoue et al. describing CD9's function in membrane compartmentalization. doi: 10.1242/dev.189985.

Response: We thank the reviewer for pointing out this oversight. We have added the requested reference as Reference 26, and have altered the manuscript to include mention of its key finding that CD9 promotes the (Line 42-43) “. . . proper localization of oocyte-surface proteins . . .”.

b. In line 63, include work by Noda et al. on DCST1/2. doi: 10.1038/s42003-022-03289-w.

Response: The requested reference has been added as Reference 35.

4. In addition to biolayer interferometry, the authors perform pulldowns to assess whether SPACA6 and IZUMO1 interact through their ectodomain and show that SPACA6 was not precipitated with recombinant IZUMO1. An important control to add is a negative control to show that the His tag does not lead to non-specific interactions.

Response: We agree with the reviewer that a bait-absent control is vital for confirming that interactions observed in pull-downs are between the target and the bait, instead of between the target and the resin. However, since our experiments revealed no interaction between SPACA6 and the bait proteins IZUMO1 or JUNO, there isn't a strong need to differentiate between specific and non-specific interactions when no interactions are observed in the experiment. There is likely no non-specific interactions by SPACA6.

5. In line 7, the claim that all proteins in the IST superfamily are involved in gamete fusion is not accurate. The other members of the IZUMO family have not been shown to be essential for gamete fusion.

Response: We agree that the other IZUMO family members have yet to be shown as essential for gamete fusion, though IZUMO3 plays a role in acrosome biogenesis and the IZUMO proteins in general have been seen to form complexes. To avoid over-stating their relevance to gamete fusion, we have modified the language in:

Abstract (Line 7) by switching “gamete fusion-associated” to “fertilization-associated”

Introduction (Line 72-73): “. . . including several linked to gamete fusion and/or fertilization”

Results (Line 255-256): removed “gamete fusion-associated” from the sentence to now read “. . . the founding members of a conserved structural superfamily that includes TMEM95 and IZUMO proteins 2, 3, and 4”.

Figure text: Changed the figure legend title to “Figure 3. SPACA6 and IZUMO1 are founding members of a fertilization-associated superfamily.”

6. The Discussion could benefit from an in-depth analysis of the functional sites of SPACA6 and their potential functional roles.

Response: We have expanded our Discussion of SPACA6's potential to form protein-protein interactions by including more in-depth analysis of each conserved patch (Lines 383-396): “The structure of SPACA6 reveals several conserved surfaces that could participate in protein-protein interactions. **The conserved portion of hinge region directly adjacent to the CXXC motifs (denoted as Patch 1 above) possesses several outward facing aromatic residues, which are commonly associated with hydrophobic and π -**

stacking interactions between biomolecules. The wide edge of the Ig-like domain (Patch 2) forms a positively charged trough with highly conserved Arg and His residues, and an antibody against this region was previously used to block gamete fusion³⁰. The antibody recognizes the linear epitope ²¹²RIRPAQLTHRGTF²²⁵, which possesses three of the six arginine residues and the highly conserved His220. Whether the function disruption resulted from occluding these specific residues or the region as a whole is unclear. The position of this cleft near the C-terminal end of the β -sandwich would suggest a *cis*-interaction with an adjacent sperm protein rather than an interaction with an oocyte protein. **In addition, the conservation of the highly flexible proline-rich coil within the hinge (Patch 3) could be a protein-protein interaction site, or – perhaps more likely – indication of a conserved flexibility between the two domains important for the yet unknown role of SPACA6 in gamete fusion.”**

Reviewer #3:

1. The author showed that SPACA6 cannot interact with JUNO. It is better to provide a structural explanation for such phenomenon. For example, are the interface or key residues that mediate the interaction between IZUMO1 and JUNO conserved in SPACA6?

Response: We thank the reviewer for their suggestion and interest in the structural reasoning behind SPACA6's inability to bind Juno. We have provided two supplementary panels in Figure S14 (panels F and G) as well as the following Results text (Line 288-296): “Despite the structural similarities between SPACA6 and IZUMO1, the former's inability to bind JUNO is not entirely surprising. There are over 20 residues on the surface of IZUMO1 that interact with JUNO, including residues from each of the three regions (though the majority are found in the hinge region) (Figure S14F). Of these residues, only one is conserved in SPACA6 (Glu70). While many residues substitutions maintain the original biochemical attributes, the essential Arg160 residue in IZUMO1 is changed to a negatively charged Asp148 in SPACA6; previous studies showed that an Arg160Glu mutation in IZUMO1 almost completely abolished binding to JUNO⁴³. In addition, the differences in domain orientation between IZUMO1 and SPACA6 drastically augment the surface shape of the putative JUNO binding site (Figure S14G).”

2. It is pity that the potential SPACA6 interacting ovum protein remains unknown. Since the structure of SPACA6 is similar to IZUMO1, it is not surprising that the SPACA6 interacting protein may be structurally similar to JUNO. Have the authors also performed a DALI search for JUNO? It would be nice to have more discussion on such perspectives in searching for the binding partners of SPACA6.

Response: We thank the reviewer for their suggestion and share their wish to discover SPACA6's oocyte binding partner (if one exists). We have run DALI searches for JUNO and received many hits. However, since JUNO shares a similar fold to functioning folate receptors throughout the tree of life, DALI searches alone cannot inform us if any of these proteins are functioning similarly to JUNO in addition to their known functions as folate receptors.

3. It is pleased to see that the authors further studied the conformational dynamic of SPACA6 using SAXS and H-DXMS methods. One wonder whether such conformational dynamic has any functional relevance?

Response: We share the reviewer's excitement and curiosity concerning the conformational dynamics of SPACA6 and its functional relevance. While our study does not provide enough evidence to definitively confirm a functional role for SPACA6 flexibility, we have added to the following sentence to share this

possibility (Line 340-343): “Interestingly, the C-terminal portions of hinge section in SPACA6, comprised of residues ¹⁴⁷CDLPLDCP¹⁵⁴, is the highly conserved Patch 3 (Figure 6B), **perhaps indicating that inter-domain flexibility is an evolutionary conserved trait in SPACA6.**”

4. Is the recombinant expressed SAPCA6 glycosylated? Does the author observe the glycosylation site in the structure?

Response: While not strong enough to build into, weak electron density at the tip of Asn243 does imply that this is, indeed, an N-linked glycosylation site. To make this apparent in the manuscript, we have added the following sentence to the structure description (Line 100-101): “Weak electron density at the end of the Asn243 sidechain suggests this residue is N-linked glycosylated.”

5. The manuscript contains too many figures, some of which can be obviously combined (e.g. S14 and S15).

Response: To condense the supplementary information, Figures S8 and S9 have been combined into Figure S8. Additionally, Figures S14 and S15 have been combined into Figure S14.

6. The structural difference seen in S10 between SPACA6 and IZUMO1 cannot be recognized as structural flexibility as they are from two different targets.

Response: Agreed. The title for Figure S10 (now Figure S9) has been changed to “Differences in domain orientation between IZUMO1 and SPACA6.”

Reviewer #1 (Remarks to the Author):

The authors have presented the structure of ectodomain of SPACA6. It is pretty clear that important functional aspects of SPACA6 may depend on transmembrane helices or linkers. Adding this detail about solving ectodomain structure in manuscript title and abstract could be beneficial.

Reviewer #2 (Remarks to the Author):

The manuscript by Vance et al. has now been significantly improved. I only have 2 minor comments:

- 1) In addition to the structural overlay in Figure S10, it would be useful for the readers to evaluate the similarity by reporting a structural similarity score.
- 2) Figure 4 would be more informative if it was replaced by an actual taxonomic or phylogenetic tree.

Reviewer #3 (Remarks to the Author):

In the revised manuscript, the authors have fully addressed my previous questions as well as the comments from other reviewers. It now reaches, in my eyes, the sufficient level for for publication.

Point-by-Point Response

Reviewer #1:

1. The authors have presented the structure of ectodomain of SPACA6. It is pretty clear that important functional aspects of SPACA6 may depend on transmembrane helices or linkers. Adding this detail about solving ectodomain structure in manuscript title and abstract could be beneficial.

Response:

We thank reviewer 1 for their comment and agree. The requested clarification concerning our solution of the ectodomain of SPACA6 has been added to both the title – which now reads “SPACA6 **ectodomain** structure reveals a conserved superfamily of gamete fusion-associated proteins” – and to the abstract – “We elucidated the crystal structure of the SPACA6 **ectodomain** at 2.2-Å resolution”.

Reviewer #2:

The manuscript by Vance *et al.* has now been significantly improved. I only have 2 minor comments:

1. In addition to the structural overlay in Figure S10, it would be useful for the readers to evaluate the similarity by reporting a structural similarity score.

Response:

We thank reviewer 2 for their continued feedback. An RMSD value of 1.1 Å was reported in the manuscript text. For clarification, this value has also been added to the caption of Figure S10: “Structures align with an RMSD of ~1.1 Å (aligning 1494 atoms).”

2. Figure 4 would be more informative if it was replaced by an actual taxonomic or phylogenetic tree.

Response:

We thank reviewer 2 for their comment concerning Figure 4. Figure 4 is intended to show the distribution of the different fusion-associated proteins throughout the disparate portions of vertebrate life. We are of the opinion that the Figure as it stands achieves this goal, and that remaking it as a phylogenetic tree would add complexity and dilute the main point. Since Figure 4 looks at the distribution of 10 different proteins, multiple phylogenetic trees would be required – one for each protein – and the absence of proteins from certain branches of the tree of life would have to be gleaned by that branch’s absence from the phylogenetic tree. Considering the reviewer frames this suggestion as a minor concern and no other reviewer took issue with the figure, we ask that Figure 4 remain as it is, thereby conserving the clarity of the figure’s point.

Reviewer #3:

In the revised manuscript, the authors have fully addressed my previous questions as well as the comments from other reviewers. It now reaches, in my eyes, the sufficient level for publication.